# Preoperative Serum Alkaline Phosphatase and Neurological Outcome of Cerebrovascular Surgery

**DOI:** 10.3390/jcm11112981

**Published:** 2022-05-25

**Authors:** Seongjoo Park, Eun-Su Choi, Hee-Won Jung, Ji-Youn Lee, Jin-Woo Park, Jae-Seung Bang, Yeong-Tae Jeon

**Affiliations:** 1Department of Anaesthesiology and Pain Medicine, Korea University Guro Hospital, Korea University College of Medicine, Seoul 08308, Korea; cardiac.anesthesiologist@gmail.com; 2Department of Anaesthesiology and Pain Medicine, Korea University Ansan Hospital, Korea University College of Medicine, Ansan 15355, Korea; potterydoll@hanmail.net (E.-S.C.); jhwish21@gmail.com (H.-W.J.); 3Department of Anesthesiology and Pain Medicine, Seoul National University Bundang Hospital, Seongnam 13620, Korea; heykiki1@hanmail.net (J.-Y.L.); jinul8282@gmail.com (J.-W.P.); 4Department of Neurosurgery, Seoul National University Bundang Hospital, Seongnam 13620, Korea; nsbang@snubh.org; 5Department of Anesthesiology and Pain Medicine, Seoul National University College of Medicine, Seoul 03080, Korea

**Keywords:** alkaline phosphatase, cerebrovascular disease, vascular surgery, predictors, risk factors

## Abstract

This study evaluated the relationship between the preoperative alkaline phosphatase (ALP) level and major postoperative neurological complications in patients undergoing cerebral bypass surgery. This was a retrospective analysis of a prospective database of all patients undergoing cerebral bypass surgery after a diagnosis of cerebrovascular stenosis or occlusion between May 2003 and August 2017. The patients were divided into tertiles based on serum alkaline phosphatase (ALP) levels (low: <63, intermediate: 63~79, and high: ALP > 79 IU/mL). The incidence of neurological events according to ALP level was analyzed. The study analyzed 211 cases. The incidence of acute infarction was highest in the third serum ALP tertile (5.7% vs. 2.9% vs. 16.9% in the first, second, and third tertile, respectively, *p* = 0.007). Logistic regression analysis showed that the third tertile of serum ALP was an independent predictor of acute cerebral infarction (odds ratio 3.346, 95% confidence interval 1.026–10.984, *p* = 0.045). On Kaplan–Meier time-to-event curves, the incidence of acute infarction increased significantly with ALP (log rank = 0.048). Preoperative serum ALP level can be used as a biomarker to predict acute cerebral infarction in patients undergoing cerebral bypass surgery for vascular stenosis or occlusion.

## 1. Introduction

Cerebral vascular bypass, including extracranial–intracranial and intracranial–intracranial bypass, is the most common surgical treatment to prevent ischemic complications in carefully selected patients with intracranial arterial stenosis, occlusion, or cerebral ischemia [1,2]. The purpose of cerebral vascular bypass is to improve blood flow in the ischemic brain area and to repair a blocked or damaged artery. However, since manipulation and temporary clipping of the artery are needed intraoperatively, postoperative complications such as subsequent stroke or hemorrhage can occur. If high-risk patients can be identified preoperatively using a reliable prognostic risk factor, it might help to guide more careful patient management, such as strict hemodynamic control, and ultimately improve the surgical outcome.

Alkaline phosphatase (ALP) is a phosphatase that releases inorganic phosphoric acid from organophosphorus esters. It is most abundant in bone and is also found in plasma. It plays an important role in bone mineralization by promoting the hydrolysis of inorganic pyrophosphate, reducing calcification inhibitors, and increasing the phosphate concentration [3]. In traditional clinical practice, blood ALP levels are used to diagnose liver or bone disease. Recently, ALP has attracted attention as a marker of vascular calcification. Increased ALP levels have been reported in patients with end-stage renal failure receiving hemodialysis [4,5,6] or with hypertension [7] or metabolic syndrome [8], and as a predictor of myocardial infarction [9,10,11]. In terms of cerebrovascular disease, ALP is a reliable predictor of the recurrence of cerebrovascular disease and mortality after stroke [12,13,14,15]. However, no study has examined the association between preoperative ALP levels and postoperative adverse neurological events in patients with cerebral ischemia undergoing surgical treatment. Therefore, this study retrospectively evaluated the relationship between preoperative ALP levels and major neurological complications in patients who underwent cerebral bypass surgery after a diagnosis of cerebrovascular stenosis or occlusion.

## 2. Materials and Methods

This retrospective study was approved by the institutional review board of the hospital. Data were collected from electronic medical records dating from May 2003 to August 2017. The trial was registered prior to patient enrollment at the Clinical Research Information Service (CriS; http://cris.nih.go.kr; registration number: KCT0003658). The inclusion criteria were a consecutive patient with American Society of Anesthesiologists (ASA) physical status classes I–III, over 18 years old with diagnosed cerebral stenosis or cerebral artery occlusion, who underwent cerebral bypass surgery under general anesthesia at a single tertiary hospital from June 2003 to August 2017. The informed consents of patients was waived because this was a retrospective clinical study. Patients with diseases that can increase serum ALP level such as hepatic diseases, diagnosed cancer, and chronic kidney disease were excluded from the analysis.

During the data-collection period, the surgery was performed by three surgeons according to our institutional protocol. There were two main anesthesiologists, and anesthesia was induced and maintained with a target-controlled infusion of remifentanil and propofol (Orchestra infusion pumps; Fresenius Vial, Brezins, France). The propofol and remifentanil concentrations were adjusted to maintain the intraoperative systolic arterial pressure within 20% of the baseline value. Patients received statin and anticoagulant therapy after admission. Under general anesthesia, the parietal or frontal branch of the superficial temporal artery (STA) driving was confirmed using a doppler ultrasound. Then the parietal or frontal branch of the STA was dissected and separated for use as a donor artery, according to the individual size of the patient. Furthermore, using doppler ultrasound, we selected one of the M4 branches of middle cerebral artery (MCA) in the region that showed the greatest decrease in perfusion for use as a recipient artery, but avoided the cortical artery directly supplying the infarction area. After making a small craniotomy, anastomosis between donor and recipient arteries was performed in the usual manner. After completion, blood flow was visually checked, and blood flow was also checked using transcranial doppler ultrasonography and indocyanine green angiogram. After surgery, hypotension or dehydration was avoided and a systolic blood pressure of >120 mmHg was maintained. Aspirin was administered at 100 mg/day from the day after surgery.

Demographic data were recorded, including age, sex, height, weight, body mass index (BMI), diagnosed disease, surgical procedure performed, underlying conditions (e.g., smoking, hypertension, diabetes, and previous stroke or myocardial infarction), and preoperative anticonvulsant infusion. Preoperative laboratory data were also collected, including hemoglobin, C-reactive protein, cholesterol, calcium, phosphorus, creatinine, albumin, bilirubin, serum aspartate aminotransferase (AST), and serum alanine aminotransferase (ALT). An adverse neurological event was postoperative acute cerebral infarction, defined as a new cerebral infarction or an increase in the size of a previous lesion that occurred within 1 month postoperatively and was confirmed by magnetic resonance imaging.

The patients were classified into three groups by tertiles based on preoperative serum ALP levels: low (ALP < 63 IU/mL), intermediate (ALP 63~79 IU/mL), and high (ALP > 79 IU/mL). All parameters were compared among these three groups. Values are described as the mean ± standard deviation or number (percent).

The statistical analysis was performed using SPSS Statistics ver. 21.0 (IBM SPSS, Chicago, IL, USA). Continuous variables were compared using one-way analysis of variance (ANOVA) or the Kruskal–Wallis test. Differences in proportions were compared using the chi-square test or Fisher’s exact test. Logistic regression analysis was performed to evaluate predictors of postoperative 30-day neurologic event. The incidence rate of acute cerebral infarction was estimated using the Kaplan–Meier product-limit estimation method with the log-rank test. After checking for violation of the proportional hazard assumption, Cox proportional hazards regression models were used to estimate the relationship between the ALP levels and adverse neurological events and acute infarction. A *p* value < 0.05 was considered statistically significant.

## 3. Results

This study enrolled and analyzed 211 patients (Figure 1). Table 1 presents the baseline demographic characteristics of the three groups based on ALP levels. Except for ALP and phosphorus levels, the demographic data, medial history, and laboratory data did not differ significantly among the three groups. In all three groups, the proportion of males was relatively high, but there was no significant difference in sex among the three groups. The ALP level differed significantly among the three groups (*p* < 0.001). Phosphorus, like ALP, differed significantly among the groups, and patients in the high ALP group tended to have higher phosphorus levels (*p* < 0.001).

The total incidences of acute cerebral infarction were 8.5%. The incidence of acute cerebral infarction in each group differed significantly among the three groups (*p* = 0.007, Table 2). Based on the occurrence of 30-day cerebral infarction, we categorized patients into two groups as follows: infarction (*n* = 18) and no infarction (*n* = 193) (Table 3). No significant intergroup differences were observed in baseline characteristics except preoperative ALP level. (72.5 ± 27.9 vs. 92.5 ± 33.3; no infarction group vs infarction group, respectively, *p* = 0.005, Table 3).

In the logistic regression analysis, only the serum ALP tertile was related to the incidence of acute infarction (OR 3.356, 95% CI 1.026–10.974, *p* = 0.045, Table 4). On Kaplan–Meier time-to-event curves, all acute infarction occurred within 12 days and differed significantly among the three groups (log rank = 0.048, Figure 2). As the ALP increased, the incidence of acute infarction increased significantly. According to the Cox proportional hazards regression analysis, the hazard ratio of acute infarction was 1.013 (95% CI 1.004–1.022, *p* = 0.007, Table 5).

## 4. Discussion

This retrospective study showed that the incidence of acute infarction increased significantly with preoperative serum ALP level in patients who underwent cerebrovascular bypass surgery after a diagnosis of cerebrovascular stenosis or occlusion. The incidence of acute infarction was highest in the third tertile ALP group and high serum ALP was a predictor of postoperative acute cerebral infarction.

Previous studies found strong correlation between the serum ALP levels and the risk of early death in patients with acute ischemic stroke [16]. In the long term, serum ALP predicted 1-year mortality and recurrent vascular events [12], and all cause and vascular death even after 3 years [15]. In addition, serum ALP was also related to cerebral micro-bleeding and hemorrhagic transformation in ischemic stroke [17,18]. However, there has been no investigation of the relationship between serum ALP level and postoperative MAE after cerebrovascular surgery. To our knowledge, this is the first study to evaluate serum ALP as a predictor of adverse neurological events in patients undergoing cerebral bypass surgery after a diagnosis of cerebrovascular stenosis or occlusion.

The significant association between ALP and cerebral infarction may be attributable to several factors. First, vascular calcification is thought to be the main mechanism. ALP accelerates the hydrolysis of pyrophosphate and potentially reduces this calcification inhibitor [3,19], which results in vascular calcification [20,21]. Consequently, ALP activity is often used as a molecular marker for vascular calcification [22,23]. Vascular calcification induces atherosclerosis, increases vascular stiffness, and decreases vascular compliance. We believe that vascular calcification causes cerebrovascular micro-stenosis in the brain, which increased susceptibility to ischemia while manipulating and temporarily clipping the artery during surgery. Second, an elevated serum ALP might reflect inflammation [24,25,26], because increased ALP levels have been reported in sepsis [27] or with elevated C-reactive protein [28]. Inflammation increases neutrophil infiltration in the brain cortex, disrupts the blood-brain barrier, reduces tissue reperfusion, and causes microvascular coagulation and complement-dependent brain injury [29]. This may lead to adverse neurological events after bypass surgery, such as hemorrhage or subsequent acute infarction. Therefore, an elevated ALP level may be a risk factor for acute infarction due to progressive atherosclerosis, as well as a risk factor for hemorrhage due to immature, unstable vascular structures. Third, damaged vascular homeostasis is another possible mechanism. The bone-type ALP affects osteoblasts [3,30], which regulate the production of hematopoietic stem cells derived from bone marrow [31,32,33], which in turn play a major role in vascular homeostasis by participating in the pathogenesis of atherosclerosis [34] and promoting angiogenesis [31]. Therefore, elevated ALP levels might constitute a risk factor for ischemic and hemorrhagic stroke.

In this study, phosphorus concentration was elevated with ALP. This is thought to occur because phosphate upregulates ALP activity [35]. This is similar to the results of studies that have investigated the correlation between phosphorus and cardiovascular disease or stroke [36,37,38,39]. In those investigations, elevated phosphorus accompanied by elevated ALP was related to the occurrence of stroke, which is related to the vascular calcification mechanism. However, phosphorus did not show predictive value in the logistic regression analysis.

Our study had several limitations. First, the number of patients was small compared with studies of myocardial infarction or stroke. However, regardless of the small number, this study clearly showed a relationship between a high serum ALP level and postoperative adverse neurological events. Further prospective, randomized studies of the effects of serum ALP on the neurological outcome of cerebrovascular bypass are needed. Second, three surgeons were involved in this study. Although we did not compare the differences in the incidences of complications according to surgeons, all three were experts with more than 5 years of experience.

## 5. Conclusions

In conclusion, preoperative serum ALP level was an independent predictor of acute infarction in patients undergoing cerebral bypass surgery. Patients with a high serum ALP may require more careful patient management to prevent postoperative complications and improve outcomes.

## Figures and Tables

**Figure 1 jcm-11-02981-f001:**
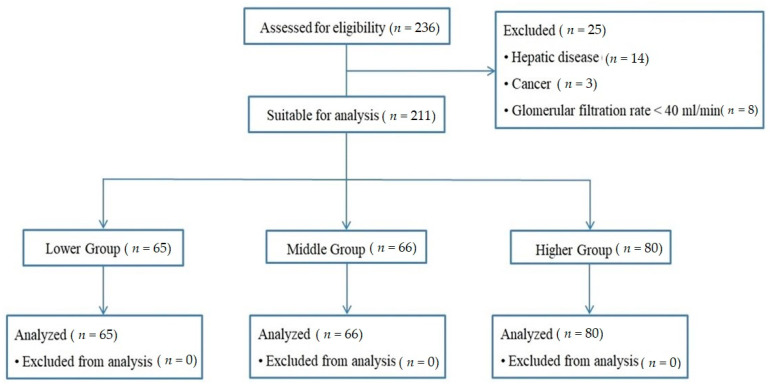
Consort flow diagram.

**Figure 2 jcm-11-02981-f002:**
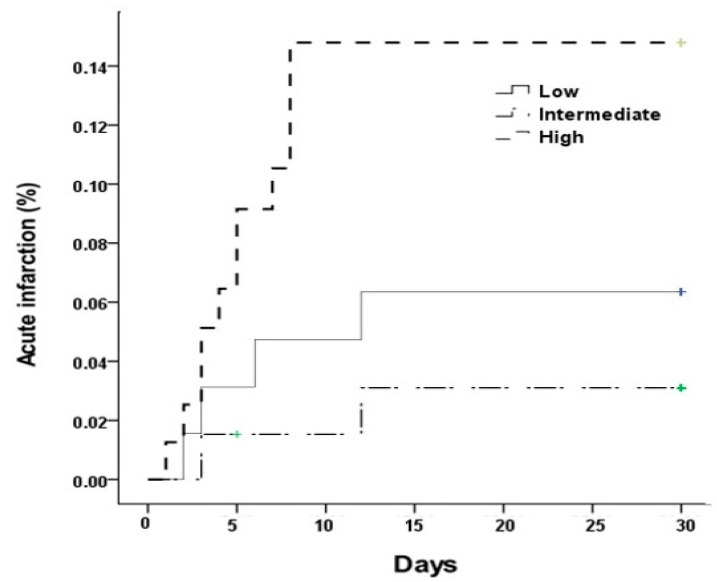
Kaplan–Meier event curves of acute cerebral infarction (Log rank 0.048).

**Table 1 jcm-11-02981-t001:** Demographic data according to the level of ALP.

	Lower Group (*n* = 70)	Middle Group (*n* = 70)	Higher Group (*n* = 71)	*p* Value
Demographic data				
Age (year)	61.8 ± 11.3	64.2 ± 11.4	64.6 ± 11.7	0.285
Body-mass index	24.0 ± 3.4	24.2 ± 2.6	24.4 ± 3.6	0.783
Male	47 (72.3%)	46 (69.7%)	58 (72.5%)	0.073
Past history				
Smoking	29 (44.6%)	23 (34.8%)	36 (45.0%)	0.371
Diabetes mellitus	25 (38.5%)	28 (42.4%)	22 (27.5%)	0.162
Hypertension	44 (67.7%)	45 (68.2%)	50 (62.5%)	0.787
Stroke	25 (38.5%)	25 (37.9%)	35 (43.8%)	0.723
Myocardial infarction	7 (10.8%)	6 (9.1%)	10 (12.5%)	0.805
Laboratory variables				
CRP (mg/L)	0.1 ± 0.2	0.3 ± 1.3	0.5 ± 1.9	0.114
Hemoglobin (g/dL)	13.5 ± 1.5	14.4 ± 7.4	13.6 ± 1.6	0.438
Cholesterol (mg/dL)	150.3 ± 44.5	158.9 ± 44.9	158.3 ± 40.1	0.427
Calcium (mg/dL)	8.7 ± 0.8	8.9 ± 0.5	8.8 ± 0.7	0.475
Phosphorus (mg/dL)	44.7 ± 17.3	60.6 ± 23.9	89.2 ± 45.0	<0.001
Creatinine	0.9 ± 0.3	1.0 ± 0.3	0.9 ± 0.3	0.273
Albumin (g/dL)	4.0 ± 0.6	4.2 ± 0.5	4.1 ± 0.4	0.238
Bilirubin (mg/dL)	0.7 ± 0.2	0.7 ± 0.3	0.6 ± 0.3	0.651
AST (IU/dL)	24.8 ± 19.1	25.5 ± 23.6	28.1 ± 14.7	0.583
ALT (IU/dL)	26.7 ± 22.8	30.8 ± 33.0	35.0 ± 25.0	0.193
ALP (IU/dL)	49.4 ± 10.6	70.2 ± 4.8	102.7 ± 30.1	<0.001

The values are presented as number (%) or mean ± standard deviation. Abbreviations: CRP: C-reactive protein; AST: serum aspartate aminotransferase; ALT: serum alanine transaminase: ALP: Alkaline phosphatase.

**Table 2 jcm-11-02981-t002:** Incidence rates of acute infarction.

	Lower Group (*n* = 70)	Middle Group (*n* = 70)	Higher Group (*n* = 71)	*p* Value
Acute infarction	4 (5.7%)	2 (2.9%)	12 (16.9%)	0.007

The values are presented as number (%).

**Table 3 jcm-11-02981-t003:** Intergroup comparison of baseline characteristics based on the incidence of 30-day acute cerebral infarction.

	No Infarction (*n* = 193)	Infarction (*n* = 18)	*p* Value
Demographic data			
Age (year)	63.7 ± 11.4	61.8 ± 12.5	0.497
Body-mass index	24.2 ± 3.3	24.1 ± 2.3	0.836
Male	136 (70%)	15(83%)	0.290
Past history			
Smoking	80 (41%)	8 (44%)	0.799
Diabetes mellitus	71 (37%)	4 (22%)	0.307
Hypertension	129 (67%)	10 (56%)	0.503
Stroke	48 (25%)	3 (17%)	0.575
Myocardial infarction	21 (11%)	2 (11%)	1.000
Laboratory variables			
CRP (mg/L)	0.3 ± 1.4	0.3 ± 0.7	0.960
Hemoglobin (g/dL)	13.9 ± 4.6	13.5 ± 2.0	0.689
Cholesterol (mg/dL)	156.7 ± 43.1	147.2 ± 44.4	0.374
Calcium (mg/dL)	8.8 ± 0.7	8.8 ± 4.8	0.969
Phosphorus (mg/dL)	64.1 ± 34.2	73.8 ± 53.3	0.280
Creatinine	0.9 ± 0.3	0.9 ± 0.2	0.775
Albumin (g/dL)	4.1 ± 0.5	4.2 ± 0.4	0.615
Bilirubin (mg/dL)	0.7 ± 0.3	0.7 ± 0.3	0.757
AST (IU/dL)	26.4 ± 20.1	23.1 ± 9.5	0.488
ALT (IU/dL)	30.8 ± 27.5	31.7 ± 27.3	0.897
AST/ALT ratio	1.0 ± 0.5	1.0 ± 0.7	0.972
ALP (IU/dL)	72.5 ± 27.9	92.5 ± 33.3	0.005

The values are presented as number (%) or mean ± standard deviation. Abbreviations: CRP: C-reactive protein; AST: serum aspartate aminotransferase; ALT: serum alanine transaminase; ALP: Alkaline phosphatase.

**Table 4 jcm-11-02981-t004:** Logistic regression analysis of the relationship between ALP and adverse neurologic event or acute infarction.

	OR (95% CI)	*p* Value
ALP third tertile	3.356 (1.026–10.974)	0.045
Cholesterol	0.995 (0.983–1.006)	0.372
AST/ALT ratio	0.983 (0.377–2.560)	0.971
ALP third tertile	3.356 (1.026–10.974)	0.045

The values are presented as number (%). Abbreviations: AST: serum aspartate aminotransferase: ALT: serum alanine transaminase; ALP: Alkaline phosphatase.

**Table 5 jcm-11-02981-t005:** Relationship between ALP and acute infarction.

	Event Number	Hazard Ratio	95% CI of Hazard Ratio	*p* Value
Acute infarction	18	1.013	1.004–1.022	0.007

## Data Availability

The data that support the findings of this study are available from the corresponding author upon reasonable request.

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
