# Peer review of "Preoperative Serum Alkaline Phosphatase and Neurological Outcome of Cerebrovascular Surgery"

_jcm, 2022, doi:10.3390/jcm11112981_

Round 1

Reviewer 1 Report

Thanks. Interesting data and conclusions. I think the results are interesting due to the topic. 

I have some thoughts?

Material and Metods

1. It is a lack of information regarding if patients were treated by statins blocks? Include that data.

2. Some of the authors is Surgeons who have performed procedures (patients) who is included I the datamaterial, correct? If so, I suggesting a statement that procedures (data material) was included from different surgeons if so. If not from all surgeons do a statement regarding that as well. 

Statistics,

3. Was there any regression model done on group level?

If not, I suggest one.

4. Was there any other output of risk factors showed from the regression model?

Author Response

First of all, I, along with my coauthors, thank the editors and reviewers who reviewed the paper.

As you mentioned, there is a lack of information regarding if patients were treated by statins blocks. All patients, regardless of group, received statin after admission, and this information was inserted into Line 79.  

Next, the surgery was performed by a total of 3 surgeons, and all were performed in the same way according to the conventional method of our institute. The surgical procedure was described into lines 80-91.

Third, logistic regression analysis was performed to evaluate predictors of postoperative 30-day neurologic event. Incidence rate of acute cerebral infarction was estimated using the Kaplan–Meier product-limit estimation method with the log-rank test. After checking for violation of the proportional hazard assumption, Cox proportional hazards regression models were used to estimate the relationship between the ALP levels and adverse neuro-logical events and acute infarction.

In the logistic regression analysis, only serum ALP tertile was related to the incidence of acute infarction. As the ALP increased, the incidence of acute infarction increased significantly. According to the Cox proportional hazards regression analysis, the hazard ratio of acute infarction was 1.013.

Threrfore, only ALP was risk factor of acute cerebral infarction.

Once again, thank you for giving your opinion so that we can develop our thesis,
and thank you for accepting our manuscript. 

Reviewer 2 Report

The authors in a retrospective study examined preoperative alkaline phosphatase (ALP) values and outcomes of patients undergoing cerebral vascular bypass  surgery.  They demonstrated that preoperative ALP serum levels are biomarkers to predict acute cerebral infarctions associated with surgery for vascular stenosis or occlusion.

The manuscript is well-written and concise.  For me, and I am sure many of the readers of this journal, the information is new.

I have only minor suggestions:

In the introduction (L35)I would clarify by inserting vascular between cerebral and bypass.  With respect to infarction, I would clarify by adding "cerebral" before infarction (L119).  Most anesthesiologists will see infarction as a cardiac event especially since Table 1 above lists myocardial infarction. 

Author Response

First of all, I, along with my coauthors, thank the editors and reviewers who reviewed the paper.

As you mentioned,  'vascular' was added to lines 35 and 38, and 'cerebral' was added to lines 98, 110, 131, and 132.

Major spelling errors have been corrected. (line 160, 165, and 238)

Figure 2 was found to be missing, so Figure 2 was added. (line 157)

Once again, thank you for giving your opinion so that we can develop our thesis,
and thank you for accepting our manuscript.

Sincerely,